# Ruthenium(II) Complex with 8-Hydroxyquinoline Exhibits Antitumor Activity in Breast Cancer Cell Lines

**DOI:** 10.3390/cancers17020195

**Published:** 2025-01-09

**Authors:** Amr Khalifa, Salah A. Sheweita, Asmaa Namatalla, Mohamed A. Khalifa, Alessio Nencioni, Ahmed S. Sultan

**Affiliations:** 1Department of Internal Medicine and Medical Specialties, University of Genoa, Viale Benedetto XV 6, 16132 Genoa, Italy; asmaa.namatalla@edu.unige.it (A.N.); alessio.nencioni@unige.it (A.N.); 2Department of Biotechnology, Institute of Graduate Studies and Research, Alexandria University, Alexandria P.O. Box 21526, Egypt; 3Department of Clinical Biochemistry, Faculty of Medicine, King Khalid University, Abha 62521, Saudi Arabia; 4Department of Chemistry, Faculty of Science, Alexandria University, Alexandria P.O. Box 21511, Egypt; ma.khalifa@alexu.edu.eg; 5Ospedale Policlinico San Martino IRCCS, Largo Rosanna Benzi 10, 16132 Genoa, Italy; 6Department of Biochemistry, Faculty of Science, Alexandria University, Alexandria P.O. Box 21511, Egypt; dr_asultan@alexu.edu.eg; 7Oncology Department, Lombardi Comprehensive Cancer Center, Georgetown University Medical Center, Washington, DC 20057, USA

**Keywords:** ruthenium complex, breast cancer, apoptosis, autophagy, cell cycle

## Abstract

Breast cancer continues to pose significant treatment challenges, driving the search for new therapeutic agents with innovative mechanisms of action. Ruthenium-based compounds are emerging as promising anticancer agents due to their unique properties, which differ from those of traditional platinum-based therapies. This study investigates the effects of the ruthenium complex Bis(quinolin-8-olato)bis(triphenylphosphine)ruthenium(II) (Ru(quin)_2_) in hormone receptor-positive and triple-negative breast cancer cell models. Our results demonstrate that this compound effectively reduces cancer cell viability by inducing programmed cell death, stimulating autophagy, and causing cell cycle arrest. These findings highlight the potential of Ru(quin)_2_ as a novel therapeutic candidate for breast cancer, warranting further exploration to evaluate its in vivo efficacy and potential for clinical application.

## 1. Introduction

Breast cancer (BC) is the most frequently diagnosed cancer and continues to be the leading cause of cancer-related mortality among women globally [1,2,3]. Despite advancements in early detection and treatment, BC continues to contribute significantly to global cancer mortality, emphasizing the urgent need for more effective therapeutic strategies [4]. In Egypt, statistics from 2018 recorded approximately 134,632 newly diagnosed cancer cases and 89,042 cancer-related fatalities, with breast cancer topping the charts in terms of incidence and mortality [5]. BC ranks as the second most prevalent cancer in Egypt and the most common among females, accounting for 38.8% of all female cancer cases [6]. Moreover, it stands as the primary cause of cancer-related deaths, responsible for 29.1% of all such fatalities, amounting to a total of 6546 deaths [7]. Among females, BC prevalence is prominent across Lower, Middle, and Upper Egypt, constituting 33.8%, 26.8%, and 38.7% of fatalities, respectively [6]. Over 70% of breast cancers are classified as estrogen receptor-positive (ER+). Although endocrine therapy is effective against these hormone-receptor-positive tumors, its benefits are limited by both primary and acquired resistance [8,9]. Triple-negative breast cancer (TNBC), defined by the absence of estrogen, progesterone, and HER2 receptors, represents a particularly aggressive subtype with fewer treatment options and a higher risk of recurrence. The incidence of BC peaks in women aged 40–59 years. The global aging population, especially in Egypt, has grown from 3.4% in 1996 to 3.7% in 2006, with projections indicating it will reach 4.6 million by 2050 [10]. This trend is expected to drive a further increase in BC cases. The high prevalence of both ER+ and TNBC underscores the urgent need for improved therapeutic strategies, especially as the aging population grows and the burden of BC intensifies. This highlights the necessity for novel therapeutic approaches to tackle both ER+ and TNBC subtypes, especially in regions with a high BC incidence and mortality [11].

The discovery of cisplatin by Rosenberg and colleagues in 1960 revolutionized malignant disease treatment [12]. Although cisplatin’s antitumor properties marked a significant milestone, its high toxicity and resistance rates pose significant challenges in clinical application. Consequently, researchers turned to non-platinum metal complexes, with ruthenium (Ru) complexes emerging as promising candidates due to their diverse modes of action, distinct from the platinum compounds currently in use [13,14,15]. Compared to other emerging non-platinum agents, such as copper or gold complexes, Ru complexes exhibit unique properties, including improved targeting of cancer cells, reduced toxicity, and efficacy against drug-resistant malignancies [16]. Quinolines have attracted considerable interest, particularly 8-hydroxyquinoline (8HQ), known for its potent coordinating ability and metal recognition properties. The anticancer activity of 8HQ-based compounds has been well-documented, making it a pivotal scaffold for anticancer agents [17,18,19]. Thus, it is expected that a combination of ruthenium(II) and 8HQ in one molecule can create an active compound with a potent antitumor effects.

Autophagy, an evolutionarily conserved catabolic process, is essential for the degradation of superfluous proteins and damaged organelles [20]. This process sequesters cytoplasmic proteins and senescent organelles within vesicles, which then fuse with lysosomes to form autophagolysosomes, facilitating their degradation [21]. Autophagy can be triggered by various conditions, including hypoxia, cellular stress, and nutrient deprivation [22]. Studies suggest that targeting autophagy may represent a promising therapeutic approach in cancer treatment [23]. However, autophagy exhibits dual roles in cancer therapy, either promoting cell survival or enhancing cell death [24,25]. Emerging evidence indicates that, in certain contexts, cancer cells exploit autophagy as a survival mechanism to evade apoptosis induced by chemotherapeutic agents or γ-irradiation [26]. Several key molecular and signaling pathways, such as the autophagy-related gene (ATG) family [27] and the MAPK signaling cascade, are central to the regulation of autophagy in various cancer types [28]. Ru-based complexes have been shown to trigger both autophagy and apoptosis in cancer cells [29,30,31,32,33,34].

Apoptosis, a programmed cell death mechanism, is another crucial pathway that maintains cellular homeostasis and eliminates damaged or cancerous cells [35]. Unlike autophagy, which can sometimes act as a survival strategy, apoptosis is typically associated with cell death in response to anticancer treatments [36]. The relationship between autophagy and apoptosis, particularly in response to Ru-based therapies, remains an area of active research [37]. Some studies indicate that autophagy may function as a precursor to apoptosis, while others suggest that these processes can act independently or antagonistically, depending on the cellular context [38]. Despite the growing interest in Ru complexes as anticancer agents, the precise interplay between autophagy and apoptosis induced by these compounds is still poorly understood. Further investigation is needed to elucidate how Ru complexes orchestrate these pathways and whether modulation of autophagy could enhance the efficacy of apoptosis-driven cancer therapies.

We aimed to examine the in vitro anticancer effects of Ru(quin)_2_ on BC cell lines, focusing on the ER+ T47D and triple-negative MDA-MB-231 lines. We assessed its impact on cell proliferation, apoptosis, autophagy, and cell cycle progression in both BC subtypes. This study sheds light on the mechanisms underlying Ru(quin)_2_-induced cell death, emphasizing its potential as a therapeutic agent for both ER+ and triple-negative breast cancers.

## 2. Materials and Methods

### 2.1. Preparation of Ru(quin)_2_

The precursors for the compound were purchased from Sigma-Aldrich, Bethesda, MD, USA: RuCl_2_(PPh_3_)_3_ (catalog number 223662-1G) and quinolin-8-ol (catalog number H6878-25G). The complex was synthesized following a previously reported procedure [39]. Quinolin-8-ol (0.29 g) was placed in a 250 mL three-neck flask and deaerated. Absolute ethanol was then added, followed by sodium hydroxide (0.08 g). The reaction mixture was stirred under an argon atmosphere for 30 min. Subsequently, RuCl_2_(PPh_3_)_2_ (0.96 g) was added, and the mixture was refluxed under argon for 8 h. The resulting red crystals were filtered, washed with water and ethanol, and dried under a vacuum. A stock solution (1.2 mM) was prepared in dimethyl sulfoxide (DMSO) and stored at −20 °C.

### 2.2. Cell Lines and Reagents

T47D and MDA-MB-231 cells were obtained from the American Type Culture Collection (LGC Standards S.r.l., Milan, Italy). Cells were authenticated by DNA fingerprinting and isozyme detection. Cells were passaged for less than 6 months before their resuscitation for this study. All cell lines were routinely tested for mycoplasma contamination by a MycoAlertTM Mycoplasma Detection Kit (Lonza, Monza, Italy). Cells were cultured in RPMI1640 medium, supplemented with 10% fetal bovine serum (FBS), 100 units/mL penicillin, and 100 μg/mL streptomycin (Thermo Fisher Scientific, Milan, Italy). Both cell lines were maintained at 37 °C in a humidified atmosphere containing 5% CO_2_. RPMI1640 medium, FBS, penicillin, and streptomycin were purchased from Thermo Fisher Scientific, Milan, Italy. Sulforhodamine B (SRB) was purchased from Sigma Aldrich S.r.l., Milan, Italy.

### 2.3. Treatments

T47D and MDA-MB-231 cells were treated with Ru(quin)_2_ under specific conditions tailored to each assay. For the sulforhodamine B (SRB) assay, cells were exposed to Ru(quin)_2_ across a concentration range of 1–120 μM for 72 h to evaluate the antiproliferative effects. Morphological analysis was conducted by treating cells with Ru(quin)_2_ at concentrations of 10, 40, and 60 μM for 72 h, after which cellular changes were observed using phase-contrast microscopy. For the colony formation assay, cells were treated with 60 μM Ru(quin)_2_ for 72 h, followed by the replacement of the medium with fresh media and incubation for two weeks to allow colony growth. Apoptosis was assessed using an ELISA-based assay after treating cells with Ru(quin)_2_ at concentrations of 10, 40, and 60 μM for 24 h. Flow cytometry was performed to analyze the cell cycle in cells treated with Ru(quin)_2_ at 40 and 60 μM for 24 h. Caspase-3 activity was measured in cells treated with Ru(quin)_2_ at concentrations of 10, 40, and 60 μM for 24 h using a colorimetric assay. For immunoblotting, cells were treated with Ru(quin)_2_ at concentrations of 5, 10, 20, 40, and 60 μM for 24 h, followed by protein extraction and analysis by SDS-PAGE and Western blotting. Quantitative real-time PCR was performed on cells treated with Ru(quin)_2_ at concentrations of 10, 40, and 60 μM for 24 h, followed by RNA extraction and analysis of gene expression.

### 2.4. Sulforhodamine B (SRB) Assay

The SRB colorimetric assay was used to evaluate the antiproliferative effects of Ru(quin)_2_ across a concentration range of 1–120 µM. T47D and MDA-MB-231 cells were seeded in 96-well plates at a density of 2000 cells per well and incubated overnight at 37 °C in a humidified atmosphere of 5% CO_2_ and 95% air to allow cell adherence. The following day, the spent medium was carefully removed, and fresh culture medium containing Ru(quin)_2_ at the specified concentrations was added in triplicate. The plates were then incubated at 37 °C under the same atmospheric conditions for an additional 24 or 72 h. Following the incubation, cells were fixed by gently adding cold 50% (*w*/*v*) trichloroacetic acid (TCA) to each well, achieving a final concentration of 10% TCA, and incubating the plates at 4 °C for 20 min. The plates were then rinsed four times with tap water and left to air dry. Next, a 0.057% (*w*/*v*) SRB solution in 1% acetic acid was added to stain the cells, followed by gentle shaking for 10 min at room temperature. After staining, the SRB solution was removed, and the plates were washed four times with 1% (*v*/*v*) acetic acid and left to air dry. For dye solubilization, 100 µL of 10 mM Trizma base was added to each well, with the plates shaken for another 10 min at room temperature. Absorbance was then measured at 515 nm using a Tecan Infinite^®^ 200 PRO plate reader (Tecan, Männedorf, Switzerland) [40].

### 2.5. Morphological Analysis

Morphological observations of T47D and MDA-MB-231 cells treated with Ru(quin)_2_ were performed to identify changes induced by the treatment. The cells were exposed to increasing concentrations (10~60 μM) of Ru(quin)_2_ for 72 h, and images were captured using an inverted phase contrast microscope at 200× magnification.

### 2.6. Colony Formation Assay

One thousand T47D and MDA-MB-231 cells were seeded in 6-well plates and cultured in their respective growth media. After 24 h, the media were removed, and the cells were washed twice with PBS. Subsequently, the cells were treated with 60 µM Ru(quin)_2_ for 72 h. Following treatment, the drug-containing media were removed, and the cells were cultured in fresh media for an additional two weeks to allow colony formation. Finally, the resulting cell colonies were counted [41].

### 2.7. Apoptosis ELISA

T47D and MDA-MB-231 cells were seeded at a density of 2 × 10^4^ cells per well in 48-well plates and incubated for 24 h. The medium was then replaced with fresh medium containing 10–60 µM Ru(quin)_2_ for both T47D and MDA-MB-231 cells. After an additional 24 h incubation, apoptosis was assessed by measuring histone release from fragmented DNA using the Cell Death Detection ELISAPLUS kit (Roche Applied Science, Indianapolis, IN, USA) as previously described [42]. Briefly, this assay is based on a quantitative “sandwich enzyme immunoassay” principle, utilizing mouse monoclonal antibodies directed against DNA and histones. The assay specifically detects mono- and oligonucleosomes in the cytoplasmic fraction of cell lysates. Cells were lysed with 200 µL of lysis buffer for 30 min at room temperature. The lysate was then centrifuged for 10 min, and 200 µL of the supernatant was collected. From this, 20 µL was incubated with anti-histone biotin and anti-DNA peroxidase at room temperature for 2 h. After three washes with incubation buffer, 100 µL of substrate solution (2,2′-azino-di(3-ethylbenzthiazoline-sulfonic acid)) was added to each well and incubated for 15–20 min at room temperature. Absorbance was measured at 405 nm using an ELISA reader (Jenway Spectrophotometer, Cambridge, UK).

### 2.8. Flow Cytometric Analysis

T47D and MDA-MB-231 cells were seeded at a density of 3 × 10^5^ cells in 100 mm Petri dishes containing their regular medium and incubated for 24 h before treatment. The medium was then replaced with fresh medium containing 40 and 60 µM Ru(quin)_2_ for both T47D and MDA-MB-231 cells, followed by a 24 h incubation. The cells were then harvested by trypsinization, washed with PBS, and centrifuged at 664 g for 5 min. After centrifugation, the cells were washed again and resuspended in PBS containing 5 μg/mL RNase-A (Sigma, St. Louis, MO, USA). To assess cell permeability and label DNA, 100 μL of HFS solution (comprising 50 μg/L propidium iodide, 0.1% sodium citrate, and 0.1% Triton X-100) was added, and the cells were incubated on ice for 30 min, protected from light. Following the manufacturer’s protocol, cell cycle analysis was performed using a FACScan Flow Cytometer (Becton Dickinson, Cambridge, UK).

### 2.9. Caspase-3 Activity Assay

Caspase-3 activity was measured using a colorimetric assay kit (BioVision, Inc., Milpitas, CA, USA) according to the manufacturer’s instructions. This kit provides a rapid and effective method for detecting caspase-3 activity in cell lysates or purified samples. T47D and MDA-MB-231 cells (5 × 10^6^) were treated with 10–60 µM Ru(quin)_2_ for 24 h. The cells were lysed in 100 µL of lysis buffer containing 10 mM HEPES (pH 7.4), 2 mM EDTA, 0.1% CHAPS, 5 mM PMSF, and 5 mM DTT. Lysis was achieved through three cycles of freezing and thawing, followed by centrifugation to remove cellular debris. The supernatant was then incubated with a buffer containing 10 mM HEPES (pH 7.4), 2 mM EDTA, 0.1% CHAPS, and the caspase-3 substrate Ac-DEVD-AFC (Ac-Asp-Glu-Val-Asp-AFC) for 1 h at room temperature. The reaction was stopped by adding 1 N HCl, and absorbance was measured at 405 nm using a spectrophotometer (Jenway, Cambridge, UK) [43].

### 2.10. Immunoblotting

T47D and MDA-MB-231 cells (5 × 10^5^) were seeded in 100 mm Petri dishes containing their respective growth media. After 24 h, the media were removed, and the cells were washed with PBS. The cells were then incubated for an additional 24 h in media containing various concentrations of Ru(quin)_2_, as indicated in the figures. Following incubation, cells were rinsed and lysed using a buffer containing 50 mM Tris-HCl (pH 7.5), 150 mM NaCl, 1% Nonidet P-40, and a combination of protease inhibitors [42]. The protein content in the samples was quantified using the Bradford assay. Equal amounts of protein (35 μg) from each group were then subjected to SDS-PAGE, followed by transfer onto a PVDF membrane (Immobilon-P, Millipore S.p.A., Milan, Italy) for further analysis. The membranes were incubated overnight with primary antibodies purchased from Cell Signaling Technology, Milan, Italy: anti-cyclin D1 (#2978; 1:1000), anti-phospho-p44/42 MAPK (Erk1/2) (Thr202/Tyr204; #4377; dilution: 1:1000), anti-Erk1/2 (#9102; dilution: 1:1000), anti-LC3B (#3868; 1:1000), anti-SQSTM1/p62 (#8025; 1:1000), anti-FIP200 (#12436; 1:1000), anti-ATG13 (#13468; 1:1000), anti-CDK4 (#12790; 1:1000), and anti-CDK6 (#13331; 1:1000). Anti-Bax (#sc-20067; 1:1000), anti-β-actin (#sc-47778; 1:10,000), anti-vinculin (#sc-73614; 1:1000), anti-rabbit IgG-HRP (#sc-2357; 1:5000), and anti-mouse IgGκ BP-HRP (#sc-516102; 1:5000) were purchased from Santa Cruz Biotechnology, Italy; and anti-p21 (#MA5-14949; 1:1000) was purchased from Invitrogen, Milan, Italy. Band intensities were visualized using enhanced chemiluminescence and were quantified with Quantity One SW software version 4.6.6 (Bio-Rad Laboratories, Inc., Hercules, CA, USA) [41].

### 2.11. Quantitative Real-Time PCR (QPCR)

T47D and MDA-MB-231 cells were seeded at a density of 5 × 10^5^ cells in 100 mm Petri dishes containing their respective growth media and incubated for 24 h prior to treatment. The media were then replaced with fresh media containing 10–60 µM Ru(quin)_2_ for both cell lines, followed by an additional 24 h incubation under these conditions. Total RNA was isolated using the RNA-spin™ Total RNA Extraction Kit^®^ (iNtRON Biotechnology, Kirkland, WA, USA), and its concentration was measured spectrophotometrically with a NanoDrop 8000 (Thermo, Hudson, NH, USA). For each sample, 2 μg of RNA was reverse transcribed into complementary DNA (cDNA) using the High-Capacity cDNA Reverse Transcription Kit (Applied Biosystems, Milpitas, CA, USA). The reverse transcription reactions were carried out in a total volume of 50 μL, following the manufacturer’s recommended protocol. Following the reverse transcription reaction, the volume was adjusted to 200 μL by adding 150 μL of DNase/RNase-free water. The reverse transcription process included incubation at 25 °C for 10 min, followed by 120 min at 37 °C, 5 min at 85 °C, and a hold at 4 °C using a Mastercycler nexus GSX1 (Eppendorf, Hamburg, Germany). The quantitative PCR (qPCR) analysis was carried out in a 10 μL reaction mixture, which included GoTaq^®^ qPCR Master Mix (Promega, Madison, WI, USA), 0.3 μM of each primer, and DNase/RNase-free water. The amplification process was monitored using the QuantStudio 5 Real-Time PCR System (Applied Biosystems, Milpitas, CA, USA). The qPCR protocol consisted of an initial polymerase activation step at 95 °C for 2 min, followed by 40 cycles of denaturation at 95 °C for 15 s, and combined annealing and extension at 60 °C for 1 min. A melting curve analysis (95 °C for 15 s, 60 °C for 15 s, and 95 °C for 15 s) was subsequently performed to confirm the specificity of the PCR products. Gene-specific primers for QPCR were purchased from OriGene Technologies, Inc., Madison, WI, USA as follows: AURKB forward sequence GGAGTGCTTTGCTATGAGCTGC and reverse sequence GAGCAGTTTGGAGATGAGGTCC; ACTB forward sequence CACCATTGGCAATGAGCGGTTC and reverse sequence AGGTCTTTGCGGATGTCCACGT. Comparisons in gene expression were performed using the 2^−ΔΔCt^ method [41].

### 2.12. Statistical Analysis

Each sample was assayed in triplicate, and each experiment was repeated three times. All statistical analyses were performed using GraphPad Prism software version 8.0 (San Diego, CA, USA). Statistical comparisons were performed using a two-tailed Student’s *t*-test. A *p*-value lower than 0.05 was considered statistically significant.

## 3. Results

Inspired by these concepts, we developed a ruthenium complex incorporating 8-hydroxyquinoline (8HQ), specifically Bis(quinolin-8-olato)bis(triphenylphosphine)ruthenium(II) [Ru(quin)_2_] (Figure 1A). We previously synthesized and thoroughly characterized this compound chemically [39].

### 3.1. Ru(quin)_2_ Induced Cytotoxicity in T47D and MDA-MB-231 Cells

To investigate the anticancer effects of Ru(quin)_2_ in vitro, we examined various concentrations of Ru(quin)_2_ in ER+ T47D and TNBC MDA-MB-231 cells. Both T47D and MDA-MB-231 cells exhibited decreased cell viability after 72 h of exposure to Ru(quin)_2_ in a dose-dependent manner, as evidenced by morphological changes (Figure 1B,C) and cytotoxic activity (Figure 1D). The IC50 values of Ru(quin)_2_ were determined to be 48.3 µM for T47D cells and 45.5 µM for MDA-MB-231 cells. Treated cells showed cytoplasmic condensation, cell shrinkage, and detachment from the substrate, indicative of apoptosis induction, compared to the untreated controls. These morphological alterations were dose-dependent and consistent with apoptotic cell death (Figure 1B,C). To further confirm the antiproliferative effects of Ru(quin)_2_, we performed a colony formation assay. Ru(quin)_2_ treatment significantly reduced the number of colonies in both T47D and MDA-MB-231 cells (Figure 1E,F). Overall, our results demonstrate that Ru(quin)_2_ exhibits potent anticancer activity against both ER+ and TNBC cell lines, suggesting its potential as a possible therapeutic agent for BC.

### 3.2. Ru(quin)_2_ Induced Apoptosis in T47D and MDA-MB-231 Cells

Our data indicate that Ru(quin)_2_ significantly reduces cell viability in both T47D and MDA-MB-231 BC cells compared to untreated controls (Figure 1). Using high concentrations of 40 and 60 µM of Ru(quin)_2_ after 24 h of treatment resulted in moderate reductions in cell viability, with percentages of approximately 75.1% and 65.1% in T47D cells, and 71.5% and 62.7% in MDA-MB-231 cells, respectively (Appendix A). These concentrations as well as 5, 10, and 20 µM, applied after 24 h of treatment, were selected to investigate the biological mechanisms underlying Ru(quin)_2_ activity. The moderate reductions in cell viability ensured sufficient cell populations remained viable, enabling an accurate assessment of Ru(quin)_2_’s pharmacological effects. The decrease in cell viability observed following Ru(quin)_2_ treatment, as determined by the SRB assay, may be attributed to either cell growth arrest or cell death. To further investigate whether the reduction in viability was due to apoptosis, we performed an ELISA-based apoptosis assay that measured histone release from apoptotic cells (Figure 2A). T47D and MDA-MB-231 cells were treated with various concentrations of Ru(quin)_2_ for 24 h, and apoptosis was confirmed by increased histone release compared to untreated control cells, following the manufacturer’s protocol.

To determine whether Ru(quin)_2_-induced apoptosis in T47D and MDA-MB-231 cells involved caspase-3 activation, caspase-3 activity was measured according to the manufacturer’s instructions. Our data reveal that Ru(quin)_2_ treatment significantly increased caspase-3 activity in both T47D and MDA-MB-231 cells compared to untreated controls after 24 h (Figure 2B). Additionally, we explored the role of Aurora B kinase (AURKB), a key regulator of mitosis and apoptosis in cancer cells [42]. In our study, mRNA expression of AURKB was downregulated in both T47D and MDA-MB-231 cells following 24 h treatment with Ru(quin)_2_ (Figure 2C). A central regulator of apoptosis is BAX, a pro-apoptotic member of the Bcl-2 family that promotes mitochondrial outer membrane permeabilization, leading to cytochrome c release and subsequent caspase activation [44]. Upregulation of BAX is crucial for the initiation of apoptosis in cancer cells [45]. Our results show that Ru(quin)_2_ treatment significantly increased BAX protein levels in both T47D (Figure 2D) and MDA-MB-231 cells (Figure 2E).

These findings suggest that Ru(quin)_2_ treatment induces markers of early apoptosis, including caspase-3 activation, BAX upregulation, and downregulation of AURKB, suggesting its pro-apoptotic potential in BC cells. Together, these mechanisms underscore the potential of Ru(quin)_2_ as an anticancer agent capable of inducing apoptosis in both ER+ and TNBC cell lines.

### 3.3. Ru(quin)_2_ Triggered Autophagy in T47D and MDA-MB-231 Cells

To investigate whether autophagy contributes to the cytotoxicity of Ru(quin)_2_, we first assessed the conversion of full-length LC3-I to LC3-II, a key marker of autophagy [20], in Ru(quin)_2_-treated T47D and MDA-MB-231 BC cells. Immunoblot analysis revealed that Ru(quin)_2_ treatment increased the conversion of soluble LC3-I to lipid-bound LC3-II in a dose-dependent manner, indicating the induction of autophagy (Figure 3A,B). Additionally, escalating doses of Ru(quin)_2_ led to significant upregulation of autophagy-related proteins, such as Atg13 and FIP200, in both T47D and MDA-MB-231 cells, further confirming Ru(quin)_2_’s role in autophagic activation. However, to determine whether this increase in LC3-II levels was due to autophagy induction or a blockade of autophagic-lysosomal fusion, we evaluated the levels of SQSTM1/p62, a selective autophagy substrate. Decreased levels of SQSTM1/p62 indicate active autophagic degradation [46]. Immunoblot analysis demonstrated that Ru(quin)_2_ treatment reduced SQSTM1/p62 levels (Figure 3A,B), confirming that Ru(quin)_2_ promotes autophagic flux in both BC cell lines. Collectively, these findings demonstrate that Ru(quin)_2_ triggers autophagic degradation in BC cells.

We further explored the underlying mechanisms driving autophagy following Ru(quin)_2_ treatment in T47D and MDA-MB-231 cells. The MAPK signaling pathway is known to regulate autophagy under various stress conditions and influences key cellular functions [28,47]. As anticipated, we observed a significant increase in phosphorylated ERK1/2 (p-ERK1/2) levels in T47D and MDA-MB-231 cells treated with varying concentrations of Ru(quin)_2_ (Figure 3A,B), suggesting activation of the MAPK pathway.

### 3.4. Ru(quin)_2_ Induced Cell Cycle Arrest in T47D and MDA-MB-231 Cells

To investigate whether the antiproliferative effects of Ru(quin)_2_ were associated with cell cycle arrest, we performed flow cytometry to analyze the cell cycle distribution in T47D and MDA-MB-231 cells. After 24 h of treatment with 60 µM Ru(quin)_2_, the percentage of cells in the G0/G1 phase increased modestly from 36.1% to 44.3% in T47D cells (Figure 4A) and from 50.2% to 55.1% in MDA-MB-231 cells (Figure 4B). This increase was accompanied by a substantial decrease in the percentage of cells in the G2/M phase, from 31.3% to 11% in T47D cells (Figure 4A) and from 41% to 11.3% in MDA-MB-231 cells (Figure 4B). Notably, the S phase distribution remained largely unchanged in both cell lines (Figure 4A,B). These findings indicate that Ru(quin)_2_ may induce cell cycle arrest at the G0/G1 checkpoint, contributing to its antiproliferative effects.

To further explore the potential mechanism underlying the G0/G1 arrest, we examined the expression of key checkpoint regulators specific to the G0/G1 phase. Treatment with Ru(quin)_2_ resulted in a dose-dependent decrease in the protein levels of cyclin D1, CDK4, and CDK6, which are critical for the progression through the G1 phase and the transition into the S phase [48], in both T47D and MDA-MB-231 cells (Figure 4C,D). Additionally, we observed a notable upregulation of the CDK inhibitor p21 (Figure 4C,D), a key factor known to inhibit CDK activity and enforce arrest at the G0/G1 checkpoint. These findings strongly correlate with the observed alterations in cell cycle distribution, highlighting the pivotal role of these proteins in mediating the G0/G1 phase arrest induced by Ru(quin)_2_. Collectively, these results suggest that Ru(quin)_2_ induces G0/G1 phase cell cycle arrest in both ER+ and TNBC cells through the downregulation of cyclin D1, CDK4, and CDK6, as well as the upregulation of p21. This cell cycle blockade at the G0/G1 checkpoint may contribute to Ru(quin)_2_’s anticancer activity.

## 4. Discussion

This study provides compelling evidence that the tested ruthenium complex, Ru(quin)_2_, exhibits potent anticancer activity against BC cells, specifically ER+ T47D and triple-negative MDA-MB-231 BC cell lines. The significance of this study lies in the effectiveness of Ru(quin)_2_ in targeting two distinct subtypes of BC, a major global health concern and one of the leading causes of cancer-related mortality [4]. Our results demonstrate that Ru(quin)_2_ effectively induces cytotoxicity in BC cells and provide valuable insights into its mechanisms of action, including the induction of apoptosis, activation of autophagy, and cell cycle arrest. These findings add to the growing body of research supporting the potential of metal-based complexes as promising cancer therapeutics [49], particularly in overcoming limitations associated with conventional chemotherapies.

Ruthenium complexes have garnered significant attention as anticancer agents due to their favorable biological properties, including selective cytotoxicity and unique modes of action compared to platinum-based drugs such as cisplatin. Previous studies have shown that ruthenium complexes, including NAMI-A and KP1019, effectively target metastatic tumors with minimal side effects, laying the foundation for the development of novel ruthenium-based therapeutics [50]. The tested Ru(quin)_2_, incorporating the 8-hydroxyquinoline moiety, builds on this legacy by enhancing anticancer activity through its unique coordination chemistry [39].

Our findings reveal that Ru(quin)_2_ significantly reduces cell viability in both T47D and MDA-MB-231 BC cells in a dose-dependent manner, with IC50 values of approximately 48.3 μM in T47D cells and 45.5 μM in MDA-MB-231 cells. This decrease in cell viability, as assessed by the SRB assay, is likely due to a combination of cell growth arrest and apoptosis, supported by morphological changes and specific apoptosis assays. The induction of apoptosis was further confirmed by an increase in histone release and caspase-3 activity in Ru(quin)_2_-treated cells, which are hallmark indicators of apoptotic cell death. These results are consistent with previous studies on ruthenium-based compounds, such as NKP-1339, the only Ru (III)-based drug that has undergone phase-Ib clinical trials in CRC treatment [29], and others [30,31,32] that trigger apoptosis via activation of the mitochondrial pathway, often associated with the upregulation of pro-apoptotic proteins such as BAX and the activation of caspases.

BAX, a key pro-apoptotic protein involved in mitochondrial membrane permeabilization, plays a central role in the intrinsic pathway of apoptosis [44]. Our data show that Ru(quin)_2_ treatment significantly upregulated BAX protein expression in both T47D and MDA-MB-231 cells, suggesting that this complex promotes apoptosis via mitochondrial disruption. This upregulation of BAX, in conjunction with increased caspase-3 activity, provides strong evidence for the activation of the mitochondrial pathway of apoptosis by Ru(quin)_2_, a mechanism commonly observed in ruthenium-based anticancer agents [16,51,52].

Additionally, AURKB plays a pivotal role in apoptosis, and its inhibition may contribute to the apoptotic effects observed [53]. A crucial finding of our study is the downregulation of AURKB mRNA expression in both T47D and MDA-MB-231 cells following Ru(quin)_2_ treatment. The downregulation of AURKB mRNA expression following Ru(quin)_2_ treatment mirrors findings from studies on other ruthenium-based complexes that target AURKB to induce apoptosis [54]. These findings suggest that Ru(quin)_2_ exerts potent anticancer effects in T47D and MDA-MB-231cells by inducing apoptosis through the upregulation of BAX, activation of caspase-3, and downregulation of AURKB.

In addition to apoptosis, autophagy—a critical cellular degradation process [21]—was also activated by Ru(quin)_2_ treatment. This was evidenced by the conversion of LC3-I to LC3-II and the reduction in SQSTM1 (p62) levels, indicative of enhanced autophagic flux. The upregulation of autophagy-related proteins such as Atg13 and FIP200 supports the notion that autophagy contributes to the cytotoxic effects of Ru(quin)_2_. These findings align with previous observations of autophagy induction by another ruthenium-based compound, highlighting the potential of Ru(quin)_2_ to leverage autophagy as a therapeutic mechanism alongside apoptosis.

Additionally, Ru(quin)_2_ treatment led to cell cycle arrest at the G0/G1 phase in BC cells, a crucial checkpoint in cell division. This was accompanied by a marked decrease in cyclin D1, CDK4, and CDK6 protein expression and a corresponding increase in the CDK inhibitor p21. These molecular changes suggest that Ru(quin)_2_ disrupts the normal progression of the cell cycle, thereby inhibiting cancer cell proliferation. These effects are consistent with those reported for other ruthenium complexes that have been shown to induce cell cycle arrest in various cancer cell lines [31,32].

In addition to its pro-apoptotic role, AURKB is a critical regulator of mitosis, specifically facilitating chromosome segregation and cytokinesis [55]. Inhibition of AURKB often results in defects during mitosis, leading to cell cycle arrest at the G2/M phase and prolonged mitotic stress, which can culminate in mitotic catastrophe and apoptosis [56]. This mitotic arrest often progresses to mitotic catastrophe, further reinforcing apoptotic pathways [55]. Although Ru(quin)_2_-induced AURKB inhibition was observed in this study, it is unlikely to have directly contributed to the G0/G1 phase cell cycle arrest, as AURKB does not play a primary role in regulating the G1 phase. Instead, the observed G0/G1 arrest is more plausibly linked to the downregulation of G1-specific regulators, such as cyclin D1, CDK4, and CDK6. Nevertheless, the inhibition of AURKB by Ru(quin)_2_ appears to synergize with its other molecular effects, leading to apoptosis and demonstrating its potential as a multi-targeted agent suppressing tumor proliferation and inducing cell death.

Despite its promising anticancer effects, translating Ru(quin)_2_ into clinical applications may face challenges, including stability in biological environments, systemic toxicity, and pharmacokinetic profiling. Determining key pharmacokinetic parameters, such as in vivo half-life, absorption, distribution, metabolism, and excretion, along with the maximum tolerated dose, will be essential. Understanding these characteristics is crucial to evaluate the Ru(quin)_2_’s therapeutic potential and to optimize the most appropriate route and schedule of administration for in vivo studies

## 5. Conclusions

Overall, the results from this study position Ru(quin)_2_ as a promising candidate for further development as an anticancer agent, particularly for BC treatment. Its ability to induce multiple forms of cell death, including apoptosis and autophagy, coupled with its capacity to disrupt cell cycle progression, underscores its potential as a versatile therapeutic tool. Future studies should aim to elucidate the detailed molecular pathways underlying Ru(quin)_2_’s effects and assess its pharmacokinetics, biodistribution, and efficacy in in vivo models. In conclusion, Ru(quin)_2_ represents a potent ruthenium-based complex with significant anticancer activity against BC cells, warranting further investigation in preclinical studies and, ultimately, clinical application.

## Figures and Tables

**Figure 1 cancers-17-00195-f001:**
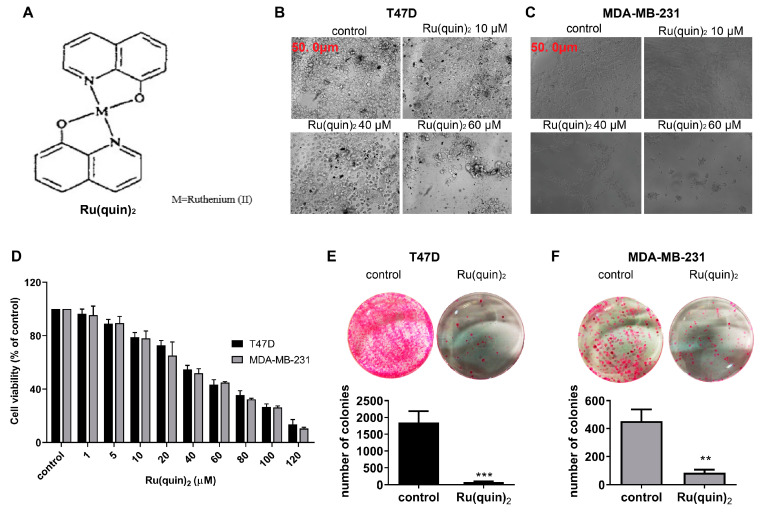
Ru(quin)_2_ triggers cytotoxicity in ER+ and TNBC breast cancer cells. (**A**) Chemical structure of Ru(quin)_2_. (**B**,**C**) Representative phase-contrast micrographs showing morphological changes in T47D (**B**) and MDA-MB-231 (**C**) cells after 72 h. (**D**) Dose-dependent inhibition of cell growth in T47D and MDA-MB-231 cells as assessed by the SRB assay after 72 h of Ru(quin)_2_ treatment. (**E**,**F**) Colony formation assay showing the inhibitory effects of Ru(quin)_2_ on T47D (**E**) and MDA-MB-231 (**F**) cells after 72 h. Colonies were stained with sulforhodamine B and quantified. Data represent the mean ± SD of three independent experiments performed in triplicate. Statistical significance: ** *p* < 0.01; *** *p* < 0.001. Abbreviations: ER+—estrogen receptor-positive, TNBC—triple-negative breast cancer.

**Figure 2 cancers-17-00195-f002:**
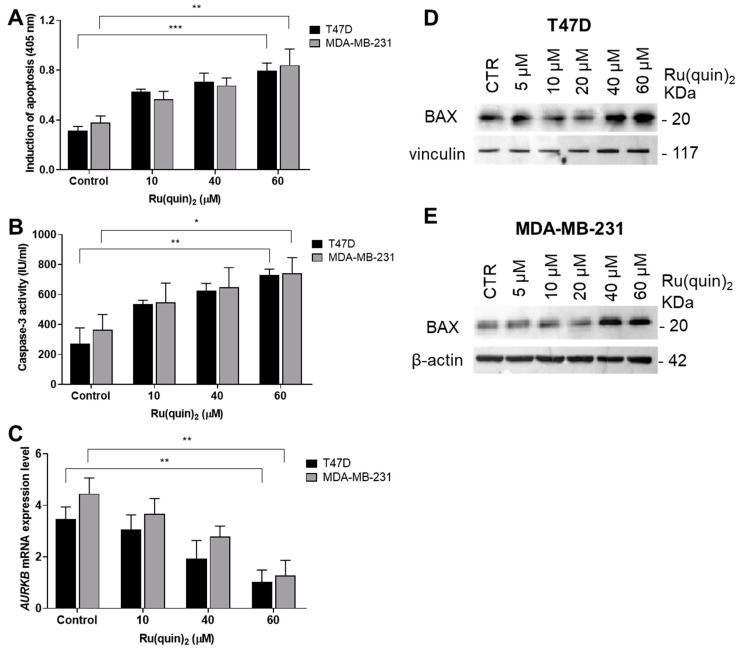
Ru(quin)_2_ induces apoptosis and alters apoptotic markers in ER+ and TNBC breast cancer cells. (**A**) Quantification of apoptosis levels in T47D and MDA-MB-231 cells using an enzyme-linked immunosorbent assay. (**B**) Caspase-3 activity assay demonstrating Ru(quin)_2_-induced caspase activation in levels of T47D and MDA-MB-231 cells. (**C**) AURKB expression levels in T47D and MDA-MB-231 cells assessed by quantitative PCR. (**D**) Western blot analysis showing the expression of BAX and vinculin in T47D cells treated with Ru(quin)_2_. (**E**) Western blot analysis showing the expression of BAX and β-actin in MDA-MB-231 cells treated with Ru(quin)_2_. Representative blot from three independent experiments is shown. Data represent the mean ± SD of three independent experiments performed in triplicate. Statistical significance: * *p* < 0.05; ** *p* < 0.01; *** *p* < 0.001. Abbreviations: ER+—estrogen receptor-positive, TNBC—triple-negative breast cancer, AURKB—Aurora B kinase. The original Western blot figures can be found in Appendix A.

**Figure 3 cancers-17-00195-f003:**
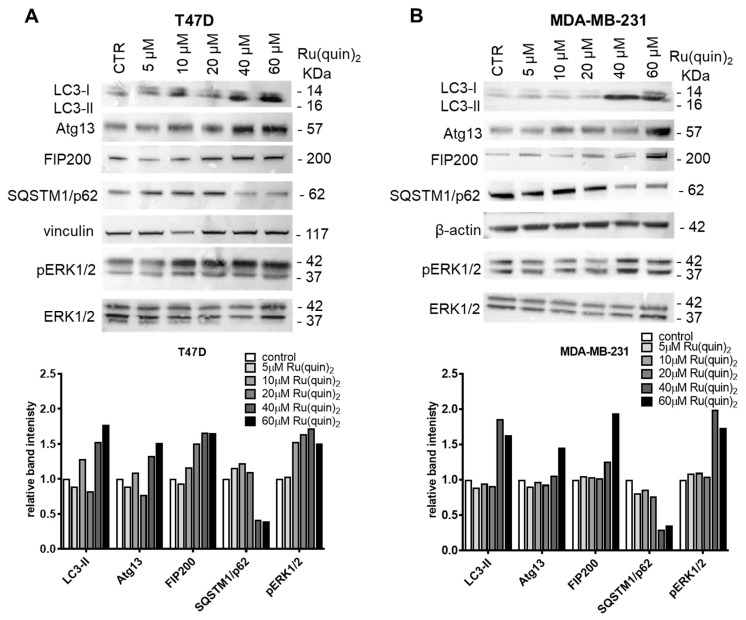
Ru(quin)_2_ promotes autophagy in ER+ and TNBC breast cancer cells. (**A**) Immunoblot analysis of autophagy-related markers LC3, Atg13, FIP200, SQSTM1/p62, vinculin, phosphorylated ERK1/2 (Thr 202 and Tyr 204), and ERK1/2 in T47D cells treated with Ru(quin)_2_. The intensity of the LC3-II, Atg13, FIP200, and SQSTM1/p62 bands was quantified and normalized to that of vinculin bands, whereas the phosphorylated ERK1/2 bands were quantified and normalized to the respective total ERK1/2 bands. (**B**) Immunoblot analysis of autophagy-related markers LC3, Atg13, FIP200, SQSTM1/p62, β-actin, phosphorylated ERK1/2 (Thr 202 and Tyr 204), and ERK1/2 in MDA-MB-231 cells treated with Ru(quin)_2_. The intensity of the LC3-II, Atg13, FIP200, and SQSTM1/p62 bands was quantified and normalized to that of β-actin bands, whereas the phosphorylated ERK1/2 bands were quantified and normalized to the respective total ERK1/2 bands. Representative blot from three independent experiments is shown. Abbreviations: ER+—estrogen receptor-positive, TNBC—triple-negative breast cancer, LC3—microtubule-associated proteins 1A/1B light chain 3, Atg13—autophagy-related protein 13, FIP200—focal adhesion kinase family interacting protein of 200 kDa, SQSTM1/p62—sequestosome 1. The original Western blot figures can be found in Appendix A.

**Figure 4 cancers-17-00195-f004:**
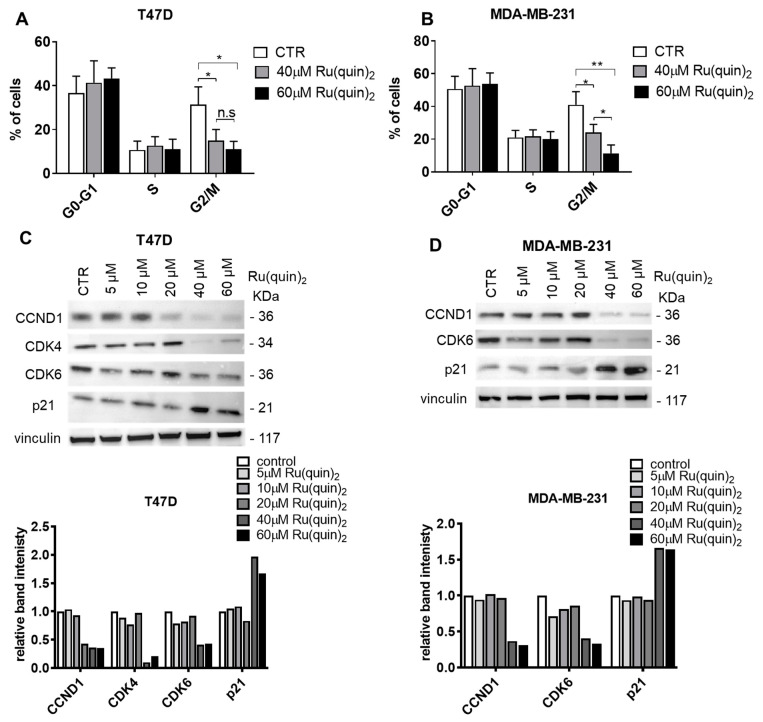
Ru(quin)_2_ induces G0/G1 phase cell cycle arrest in ER+ and TNBC breast cancer cells. (**A**) Flow cytometry analysis of the percentage of cells in each cell cycle phase (G1, S, and G2/M) in T47D cells treated with Ru(quin)_2_. (**B**) Flow cytometry analysis of cell cycle phases in MDA-MB-231 cells treated with Ru(quin)_2_. (**C**) Immunoblot analysis of CCND1, CDK4, CDK6, p21, and vinculin in T47D cells following Ru(quin)_2_ treatment. The intensity of the CCND1, CDK4, CDK6, and p21 bands was quantified and normalized to that of vinculin bands. (**D**) Immunoblot analysis of the same markers in MDA-MB-231 cells. The intensity of the CCND1, CDK6, and p21 bands was quantified and normalized to that of vinculin bands. Data represent the mean ± SD of three independent experiments performed in triplicate. Representative blot from three independent experiments is shown. Statistical significance: * *p* < 0.05; **: *p* < 0.01; n.s: non-significant. Abbreviations: ER+—estrogen receptor-positive, TNBC—triple-negative breast cancer, CDK—cyclin-dependent kinase, CCND1—cyclin D1. The original Western blot figures can be found in Appendix A.

## Data Availability

All data generated or analyzed during this study are included in this published article.

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
