# Peer review of "Ruthenium(II) Complex with 8-Hydroxyquinoline Exhibits Antitumor Activity in Breast Cancer Cell Lines"

_cancers, 2025, doi:10.3390/cancers17020195_

Round 1

Reviewer 1 Report

Comments and Suggestions for Authors

Review of the Manuscript: Ruthenium(II) complex with 8-hydroxyquinoline exhibits antitumor activity in breast cancer

The manuscript investigates the anticancer potential of a ruthenium complex, Ru(quin)â‚‚, in hormone receptor-positive and triple-negative breast cancer models. The study is well-conceived and addresses a timely topic of significant interest in cancer research. It provides a solid combination of mechanistic insights into apoptosis, autophagy, and cell cycle arrest, which supports the therapeutic promise of Ru(quin)â‚‚. The data is well-presented and convincing. Below, I provide a few minor comments that may enhance the manuscript's clarity and impact.

Minor Comments

Title” Please change “breast cancer”- into-“breast cancer cell lines” in the title.

  1. Abstract
    • The phrase "warranting further exploration to evaluate its therapeutic potential in vivo and its possible clinical application for improving treatment outcomes" is slightly repetitive. Consider condensing this for improved readability.
  1. Introduction
    • A brief mention of how Ru complexes compare with other emerging non-platinum agents in terms of mechanism or efficacy could provide a broader context.
  1. Figures and Legends
    • In Figure 1, panels B and C depicting morphological changes would benefit from including a scale bar for clarity.
    • Figure legends are generally descriptive, but explicitly stating the statistical tests used for significance in each figure would enhance rigor.
  1. Results
    • In Section 3.2, the claim “Ru(quin)â‚‚ promotes apoptosis in BC cells” could be tempered slightly by acknowledging that the study primarily demonstrates early apoptotic markers (e.g., caspase-3 activation and BAX upregulation).
  1. Discussion
    • The discussion highlights the mechanisms well but could be expanded to consider potential challenges or limitations of translating Ru(quin)â‚‚ into a clinical setting (e.g., delivery, stability, or toxicity).
  1. References
    • Some references, particularly older ones (e.g., Rosenberg’s discovery of cisplatin), may be supplemented with recent citations to demonstrate ongoing progress in the field.

General Feedback

This is a well-structured manuscript with compelling data. My comments are primarily editorial and aimed at improving the clarity and readability of the text. I do not see the need for additional experiments or major revisions. The manuscript is suitable for publication following minor revisions.

Author Response

Below please find our point-by-point reply to the comments raised by the reviewers about the research article “Ruthenium(II) Complex with 8-hydroxyquinoline Exhibits Antitumor Activity in Breast Cancer Cell lines” (cancers-3345894), by Amr Khalifa and colleagues

Reviewer #1:

Title

  1. Reviewer’s Comment: Please change “breast cancer” into “breast cancer cell lines” in the title.

We thank the reviewer for the suggestion. The title has been modified to “Ruthenium(II) Complex with 8-hydroxyquinoline Exhibits Antitumor Activity in Breast Cancer Cell lines” for improved precision.

Abstract

  1. Reviewer’s Comment: The phrase "warranting further exploration to evaluate its therapeutic potential in vivo and its possible clinical application for improving treatment outcomes" is slightly repetitive. Consider condensing this for improved readability.

We appreciate the suggestion. The sentence has been revised to: “Warranting further exploration to evaluate its in vivo efficacy and potential for clinical application.”

Introduction

  1. Reviewer’s Comment: A brief mention of how Ru complexes compare with other emerging non-platinum agents in terms of mechanism or efficacy could provide a broader context.

We thank the reviewer for the valuable input. A paragraph has been added to compare Ru complexes with other non-platinum agents, such as copper and gold complexes, to provide broader context (lines 81–84).

Figures and Legends

  1. Reviewer’s Comment: In Figure 1, panels B and C depicting morphological changes would benefit from including a scale bar for clarity.

We have added scale bars to panels B and C in Figure 1 for improved clarity as suggested.

  1. Reviewer’s Comment: Figure legends are generally descriptive, but explicitly stating the statistical tests used for significance in each figure would enhance rigor.

We thank the reviewer for this suggestion. The statistical tests have been detailed in subsection 2.12 “Statistical Analysis” of the Methods section, as recommended. This approach avoids redundancy while maintaining rigor.

Results

  1. Reviewer’s Comment: In Section 3.2, the claim “Ru(quin)â‚‚ promotes apoptosis in BC cells” could be tempered slightly by acknowledging that the study primarily demonstrates early apoptotic markers (e.g., caspase-3 activation and BAX upregulation).

We appreciate the reviewer’s observation. The sentence has been rephrased to acknowledge that the study primarily demonstrates early apoptotic markers, such as caspase-3 activation and BAX upregulation (lines 357–359).

Discussion

  1. Reviewer’s Comment: The discussion highlights the mechanisms well but could be expanded to consider potential challenges or limitations of translating Ru(quin)â‚‚ into a clinical setting (e.g., delivery, stability, or toxicity).

We thank the reviewer for highlighting this point. A paragraph discussing the potential challenges and limitations of translating Ru(quin)â‚‚ into a clinical setting, including delivery, stability, and toxicity, has been added to the Discussion section (lines 496– 502).

References

  1. Reviewer’s Comment: Some references, particularly older ones (e.g., Rosenberg’s discovery of cisplatin), may be supplemented with recent citations to demonstrate ongoing progress in the field.

We have replaced the older reference (Rosenberg, B. et al., 1978) with a more recent citation (Adhikari, S. et al., 2024) to reflect ongoing advancements in the field.

General Feedback

  1. Reviewer’s Comment: This is a well-structured manuscript with compelling data. My comments are primarily editorial and aimed at improving the clarity and readability of the text. I do not see the need for additional experiments or major revisions. The manuscript is suitable for publication following minor revisions.

We sincerely thank the reviewer for their positive assessment and constructive feedback. We have addressed all comments and implemented the suggested revisions to enhance the manuscript’s clarity and overall quality.

Thanks again for your consideration.

With my warmest regards,

Amr Khalifa, PhD

AIRC PostDoc Researcher at the Laboratory of Prof. Alessio Nencioni

Department of Internal Medicine and Medical Specialties- DIMI

University of Genoa

Viale Benedetto XV 6, 16132 Genoa, Italy

Phone: +39 010 353 8990

Email: amr.khalifa@edu.unige.it

Reviewer 2 Report

Comments and Suggestions for Authors

general comments

The study highlights the anticancer potential of Ru(quin)â‚‚ through apoptosis, autophagy, and cell cycle arrest in ER-positive and triple-negative breast cancer models. While the findings are promising, concerns about concentration-related cell viability, particularly at higher doses, raise questions about the reliability of some biological assessments.

- The IC50 values for both cell lines are approximately 45–48 µM. Subsequent biological activity assessments were conducted at concentrations around the IC50 (40 µM) and above (60 µM). At these concentrations, most cells were non-viable, making it challenging to measure the pharmacological or biological effects on living cells. Additionally, when more than half of the cells are non-viable, it becomes unclear how the compound interacts with the cells and what biological processes are occurring. Most observed effects were at the higher concentrations of 40 and 60 µM, which raises concerns about the reliability of these findings. To ensure scientifically sound results, it is recommended to maintain at least 70% cell viability when studying the biological activity of a compound, even when working with cancer cells where cell death is a desired outcome. Maintaining higher cell viability is essential to accurately assess the compound's pharmacological effects and underlying biological mechanisms.

specific comments

- Figure 1, showing morphological and colony formation results, highlights a concentration issue, particularly at 60 µM, where no cells are observed. This reinforces the importance of addressing the concentration concerns raised in the general comments.

- It would be better to move Figure 1B–F to the Results section (3.1), while keeping Figure 1A in its original position.

- It would be better to include a quantification of the immunoblot analysis in Figure 3A and B as well as Figure 4C and D, in addition to the blots, to assess their significance in terms of concentration and control.

- The results of Figure 4A and B are unreliable due to issues with the concentrations used.

Author Response

Below please find our point-by-point reply to the comments raised by the reviewers about the research article “Ruthenium(II) Complex with 8-hydroxyquinoline Exhibits Antitumor Activity in Breast Cancer Cell lines” (cancers-3345894), by Amr Khalifa and colleagues

Reviewer #2:

General Comments

Comment 1: The study highlights the anticancer potential of Ru(quin)â‚‚ through apoptosis, autophagy, and cell cycle arrest in ER-positive and triple-negative breast cancer models. While the findings are promising, concerns about concentration-related cell viability, particularly at higher doses, raise questions about the reliability of some biological assessments.

We sincerely thank the reviewer for their insightful and valuable comment, which has helped us identify and correct an important typo in our manuscript. Peer review by experts in the field is invaluable for enhancing the clarity and scientific rigor of our work. In our original submission, we mistakenly stated that the incubation period for both the cell viability and colony formation assays was 24 hours. However, the results presented in Figure 1B–F reflect 72 hours of incubation with Ru(quin)â‚‚.

We also confirm that the mechanistic and biological studies, including Western blotting, ELISA, caspase-3 activity assays, qPCR, and flow cytometry, were performed after 24 hours of incubation with Ru(quin)â‚‚. We apologize for any confusion caused by this inconsistency and have corrected the text to ensure clarity.

To further address this concern, we have included a dedicated subsection in the Materials and Methods section, titled 2.3 Treatments, where all treatment conditions, including concentrations and incubation periods, are clearly stated. We hope this addition improves the transparency of our experimental procedures.

Comment 2: The IC50 values for both cell lines are approximately 45–48 µM. Subsequent biological activity assessments were conducted at concentrations around the IC50 (40 µM) and above (60 µM). At these concentrations, most cells were non-viable, making it challenging to measure the pharmacological or biological effects on living cells. Additionally, when more than half of the cells are non-viable, it becomes unclear how the compound interacts with the cells and what biological processes are occurring. Most observed effects were at the higher concentrations of 40 and 60 µM, which raises concerns about the reliability of these findings. To ensure scientifically sound results, it is recommended to maintain at least 70% cell viability when studying the biological activity of a compound, even when working with cancer cells where cell death is a desired outcome. Maintaining higher cell viability is essential to accurately assess the compound's pharmacological effects and underlying biological mechanisms.

To address this important concern, we generated a cell viability curve using the same Ru(quin)â‚‚ concentrations presented in Figure 1D of the submitted manuscript. This curve, now included as Supplementary Figure S1, demonstrates the effect of Ru(quin)â‚‚ on cell viability after 24 hours of incubation.

Specifically, at 40 and 60 µM concentrations, cell viability percentages were approximately 75.1% and 65.1% for T47D cells, and 71.5% and 62.7% for MDA-MB-231 cells, respectively. These results confirm that the biological mechanisms investigated in our study were assessed at viability levels consistent with reliable pharmacological evaluations. We appreciate the reviewer’s comment, as it allowed us to validate and clarify the experimental conditions, ensuring robust and scientifically sound findings.

Specific Comments

Comment 3: Figure 1, showing morphological and colony formation results, highlights a concentration issue, particularly at 60 µM, where no cells are observed. This reinforces the importance of addressing the concentration concerns raised in the general comments.

We kindly refer the reviewer to our responses to General Comments 1 and 2, which address the concentration-related concerns in detail.

Comment 4: It would be better to move Figure 1B–F to the Results section (3.1), while keeping Figure 1A in its original position.

We thank the reviewer for this valuable suggestion and agree with their recommendation. As requested by the editors, we have already moved Figure 1 to the beginning of the Results section. We hope this adjustment aligns with the reviewer’s expectations.

Comment 5: It would be better to include a quantification of the immunoblot analysis in Figure 3A and B as well as Figure 4C and D, in addition to the blots, to assess their significance in terms of concentration and control.

We appreciate this insightful suggestion. We have now included quantification of the immunoblot analysis in Figure 3A and B as well as Figure 4C and D, alongside the original blots. This addition enhances the significance and interpretability of the results.

Comment 6: The results of Figure 4A and B are unreliable due to issues with the concentrations used.

As previously mentioned in General Comments 1 and 2, we have clarified the treatment conditions and provided a cell viability curve (Supplementary Material) showing that after 24 hours of treatment, cell viability remained sufficiently high in both cell lines (approximately 75%–62%) at the concentrations used. This ensures the reliability of the results presented in Figure 4A and B.

We thank the reviewer once again for their thoughtful comments, which have helped us improve the accuracy, clarity, and overall quality of our manuscript. We have addressed all concerns and implemented the suggested changes to ensure the robustness and reliability of our study.

Thanks again for your consideration.

With my warmest regards,

Amr Khalifa, PhD

AIRC PostDoc Researcher at the Laboratory of Prof. Alessio Nencioni

Department of Internal Medicine and Medical Specialties- DIMI

University of Genoa

Viale Benedetto XV 6, 16132 Genoa, Italy

Phone: +39 010 353 8990

Email: amr.khalifa@edu.unige.it
